# Electromyographic Validation of Spontaneous Facial Mimicry Detection Using Automated Facial Action Coding

**DOI:** 10.3390/s23229076

**Published:** 2023-11-09

**Authors:** Chun-Ting Hsu, Wataru Sato

**Affiliations:** Psychological Process Research Team, Guardian Robot Project, RIKEN, Soraku-gun, Kyoto 619-0288, Japan

**Keywords:** spontaneous facial mimicry, electromyography, facial action coding system, FaceReader, Py-Feat, OpenFace, cross-correlation

## Abstract

Although electromyography (EMG) remains the standard, researchers have begun using automated facial action coding system (FACS) software to evaluate spontaneous facial mimicry despite the lack of evidence of its validity. Using the facial EMG of the zygomaticus major (ZM) as a standard, we confirmed the detection of spontaneous facial mimicry in action unit 12 (AU12, lip corner puller) via an automated FACS. Participants were alternately presented with real-time model performance and prerecorded videos of dynamic facial expressions, while simultaneous ZM signal and frontal facial videos were acquired. Facial videos were estimated for AU12 using FaceReader, Py-Feat, and OpenFace. The automated FACS is less sensitive and less accurate than facial EMG, but AU12 mimicking responses were significantly correlated with ZM responses. All three software programs detected enhanced facial mimicry by live performances. The AU12 time series showed a roughly 100 to 300 ms latency relative to the ZM. Our results suggested that while the automated FACS could not replace facial EMG in mimicry detection, it could serve a purpose for large effect sizes. Researchers should be cautious with the automated FACS outputs, especially when studying clinical populations. In addition, developers should consider the EMG validation of AU estimation as a benchmark.

## 1. Introduction

Facial expressions are essential elements of nonverbal communication during social interactions [1,2,3]. In such interactions, interpersonal coordination through behavior matching and interactional synchrony [4] facilitates social alignment [5]. As a type of behavior matching, spontaneous facial mimicry of facial expressions while observing a face emerges within 1 s after stimulus onset [6,7]. Classical cases have demonstrated the use of electromyography (EMG) to record muscle-activation patterns similar to the presented face stimuli [6,8,9,10,11,12,13]. Alternatively, participants were videotaped and their facial action patterns were coded for mimicry detection using the facial action coding system (FACS) [14,15,16,17]. The FACS analyzes and describes facial movements in terms of anatomical-based action units (AUs) [18,19,20] and was designed to be manually coded by humans.

With advances in computer vision and machine learning, the FACS has been incorporated by several automatic facial behavior software, and studies have investigated the validity of such tools. Still, most of these studies have either compared automated AU-coding performances with human coder performances [21,22,23,24,25] or tested the performance of emotion recognition from labeled facial expression datasets [22,26].

A few studies directly compared automatic AU-coding performances with corresponding facial EMG data. For example, in the study by D’Arcey, participants were instructed to volitionally mimic pictures of happy and angry facial expressions as facial video and corresponding corrugator supercilii (CS) and zygomaticus major (ZM) EMG were being recorded. Significant correlations were found between FaceReader 4.0’s discrete emotion assessment and corresponding EMG activities (“happy” with ZM; “angry”, “disgusted”, and “sad” with CS) [27]. In another study by Beringer and colleagues, participants were instructed to volitionally use joyful and angry facial expressions in a response-priming task. Video for iMotion Emotient’s FACET AU4 (brow lowerer) and AU12 (lip corner puller) analysis and the corresponding CS and ZM EMG recordings were simultaneously recorded. Comparable priming effects on response time between FACET outputs and EMG recordings were found [28]. In a recent study, Kulke and colleagues compared the performance of AU4 (brow furrow) and AU12 (smile) estimates outputted by Affectiva AFFDEX [29] using EMG ZM and CS measurements when participants were asked to volitionally imitate happy, angry, and neutral facial expressions. They reported a significant correlation between the difference scores of the AFFDEX AU estimations and EMG measurements [30]. However, the detection of spontaneous facial mimicry during passive viewing was not tested.

Compared with volitional facial expressions or mimicry in the aforementioned studies, spontaneous mimicry during passive viewing tends to be subliminal for visual observation [31,32]. One could question whether it could reliably be detected via the automated FACS, whose validity has rarely been examined by research. Höfling and colleagues adopted the passive-viewing approach. Participants were presented with the International Affective Picture System scenes in one of their studies. Simultaneously, the EMG of their CS and ZM and their facial videos were recorded, which were analyzed using FaceReader 7.1. They focused on the valence and arousal estimates outputted by FaceReader. However, significant interaction was found between the stimulus category and time window in the ANOVA for EMG delta (ZM amplitude minus CS amplitude) and FaceReader AU4 and AU12, indicating that these measurements could detect spontaneous facial mimicry. FaceReader valence outputs were found to show longer latency and smaller effect sizes than EMG [33]. In another study, Höfling and colleagues tested volitional mimicry, passive viewing (spontaneous mimicry), and inhibition of mimicry conditions. While volitional mimicry could be detected by the EMG delta and FaceReader valence output, spontaneous facial mimicry during passive viewing could only be detected using the EMG delta measure, not the FaceReader valence output [34]. In this study, the performance of the FaceReader AU 4 and 12 outputs was neither estimated nor compared with corresponding EMG CS and ZM recordings. Please see Table 1 for a summary of the approaches for the FACE-EMG performance comparison between previous and present studies.

Although the validity of spontaneous facial mimicry detection has not been statistically examined, several studies have already used the automated FACS to quantitatively estimate spontaneous facial mimicry in both clinical and nonclinical populations [35,36,37,38]. The present study used the live image relay system established in a previous study [39,40]. Here, participants viewed live and videotaped dynamic facial expressions of positive (smiling) and negative (frowning) emotions with simultaneous EMG recording of the ZM and CS and frontal facial recording using the camera behind the prompter (see Materials and Methods for details). This study validates the performance of spontaneous facial mimicry detection with the automated FACS compared to the gold standard of EMG measurements. The following research questions that previous studies did not address will be covered. (1) Given the specific muscle–AU correspondence of CS–AU4 and ZM–AU12, it is possible to estimate whether the sign (positive vs. negative) of spontaneous facial response during passive viewing estimated by the automated FACS is congruent with the corresponding EMG measurements. (2) Facial mimicry requires congruent facial responses between the mimicker and the mimickee. It will be tested whether the automated FACS can detect spontaneous facial mimicry when EMG measurements do so. (3) Unlike previous studies, the correlations between muscular and estimated AU responses will be statistically tested. (4) Spontaneous facial mimicry was found to be enhanced under live face-to-face interactions compared with passive viewing of prerecorded videos [39,40]. It will be tested whether the automated FACS could also detect such modulatory effects of the live condition. (5) The exact latency of spontaneous facial mimicry detected by EMG versus the automated FACS has not been tested statistically. Cross-correlation will be used to test the extent of detection latency. (6) Previous studies validating the detection of spontaneous facial mimicry using the automated FACS employed earlier versions of commercial software, such as FaceReader 7 [33,34]. In this study, FaceReader 9.0, which incorporated updates such as deep learning-based algorithms (see Section 2.6.1) and two new open-source software, OpenFace 2.2.0 [41] and Py-Feat 0.6.0 [42], will be estimated.

Per American Psychological Association style, this study is structured as follows. Section 1 provides the literature review in relation to the present study; Section 2 provides a comprehensive overview of the methods and materials; Section 3 presents the statistical results and visualization; and Section 4 discusses the research results, limitations, and future directions.

## 2. Materials and Methods

This section presents details of the design, execution, and technical background of the analysis essential to the present study.

### 2.1. Participants

Fifty women were recruited from Kyoto and Osaka, Japan (mean ± *SD* age = 22.88 ± 2.24 years; range: 18–27 years). The data of 23 participants were previously published, which were collected between August and October 2019 [39], and the remaining ones were evaluated between August and September 2020. This study was approved by the ethics committee of the Unit for Advanced Studies of the Human Mind, Kyoto University, and was conducted in accordance with the committee’s ethical standards. Written informed consent was obtained from all participants with respect to publishing their data and being recorded on video. All of them received a monetary reward.

The simr package [43] was used to estimate the required sample size to detect the interaction effect between emotion and presentation condition based on a ZM EMG linear mixed-effect model analysis from a study that used the same paradigm [39]. The simulation demonstrated the need for more than 17 participants for the interaction effect of ZM EMG data to reach a power of 80%.

### 2.2. Facilities

Each model and participant faced a prompter. Behind the mirror of each prompter, a concealed VIXIA HF R800 camera (Canon Inc., Tokyo, Japan) was placed [39]. Videos were recorded with a full HD resolution of 1920 × 1080 pixels at 29.97 frames per second (fps). The models viewed “live-relay images” of the participants. The experiments were conducted using the Presentation program (version 20, Neurobehavioral Systems, Inc., Berkeley, CA, USA) on a Precision T3500 computer (Dell Inc., Round Rock, TX, USA) with a Windows 7 Professional operating system (Microsoft Corp., Redmond, WA, USA). Serial signals were sent to an SL-41C switcher (Imagenics Co., Ltd., Tokyo, Japan) to switch the visual input of the participant’s prompter between the computer and the model’s camera. Sounds were transmitted to the models via Presentation software using a Pasonomi TWS-X9 Bluetooth earphone (Shenzhen City Meilianfa Technology Co., Ltd., Shenzhen, China), which instructed them to produce dynamic facial expressions. The switcher maintained the resolution (1280 × 960) and the height-to-width ratio of the video output of the Presentation software and live-relay images. A BrainAmp ExG MR amplifier (Brain Products, Munich, Germany) and BrainVision Recorder software (version 2.2.1, Brain Products) were used to record EMG from participants’ ZM and CS.

### 2.3. Paradigm and Procedures

This study adopted 2 × 2 design with presentation (video vs. live) and emotion (positive vs. negative) as factors, resulting in four conditions. After written informed consent was obtained and the EMG electrodes were attached, the participant and model conversed for 3 min through the prompter. Here, it was demonstrated to the participants that the prompter system would transmit live images of the model. The conversation was followed by 8 practice passive-viewing trials, 60 actual passive-viewing trials, 4 practice-rating trials, and 16 actual-rating trials. The total testing time for the study was less than 1 h per session.

During the passive-viewing component, the participants fixated on a cross (mean intertrial interval = 2604 ms; range: 2000–3750 ms) until they were shown an announcement of either “Video” or “Real Person” for 1 s. The instruction allowed for a clear audience effect [44,45,46]. During the video trials, one prerecorded video clip was presented immediately after the instructions. During the live trials, the models performed dynamic facial expressions according to the instructions in their earphones. After showing the facial expressions, the screen displayed the fixation cross again. In the behavioral experiment, after 8 practice trials, the participants completed 15 passive-viewing trials per condition for a total of 60 trials, with a break after 32 trials (8 trials per condition). EMG data were collected during the passive-viewing trials, where no rating was requested. For each participant, the sequence of conditions during the trial was pseudo-randomized, whereas the presentation sequence of the prerecorded videos per condition was randomized. After the passive-viewing trials, participants provided 4 practice- and 16 test-subjective experiential valence and arousal ratings, during which no EMG data were acquired. Rating data were not used in the present study. The participants were remunerated for their participation and then dismissed.

### 2.4. Prerecorded and Live Videos

Two female models each recorded more than 20 videos of positive and negative dynamic facial expressions. They were instructed to display happy, smiling faces for positive expressions and angry, frowning faces with protruding mouths for negative expressions with consistent intensity. Their appearances were consistent between the video clips and live performances. An apron covered their clothes and hairpins were used to fix their hairstyles. The clips lasted 3 s, featuring a neutral expression, gradual dynamic changes, and a maximal emotional facial expression, all sustained for 1 s. Two smiling and two frowning clips were used for the positive- and negative-video conditions in the practice passive-viewing trials, respectively; 15 clips per video condition were used in the actual passive-viewing trials. During the passive-viewing trials, no clips were repeated within a single condition. The positive-live and negative-live conditions did not use prerecorded clips, and the models’ live performances were relayed.

The models’ live performances of dynamic facial expressions during the passive-viewing trials were recorded and visually inspected to ensure validity. One positive trial for two participants and one negative trial for two other participants were excluded from the analysis because they were performed either incorrectly or after a noticeable delay. Video recordings of live performances in the 2019 sessions were validated by naïve participants who could not differentiate between the prerecorded stimuli and live-performance recordings [39].

### 2.5. EMG Data Preprocessing

Raw EMG data were preprocessed using the EEGLAB MATLAB toolbox (version 2019.1; Swartz Center for Computational Neuroscience, San Diego, CA, USA). A notch filter was applied at around 60 Hz and multiples of 60 Hz; a high-pass filter was applied at 20 Hz, and a low-pass filter was applied at 500 Hz. The data were then manually screened for movement artifacts. For each trial, the signal was detrended and baseline-corrected for low-frequency drift. For this preprocessing step, the baseline was defined as the mean value from 3 s before stimulus onset to 1 s afterward, which signaled the end of the model’s neutral expression immediately before the dynamic facial expression change. All data points in each trial were shifted by the per-trial baseline mean value. The data points were then rectified to retrieve oscillation amplitudes. Absolute or rectified values (oscillation amplitudes) plus 1 were natural-log-transformed to correct for right-skewness of raw data distribution.

### 2.6. Automated AU Estimation

The participants’ face recordings were cut into 3 s clips corresponding to each trial based on the timing recorded in the log files. All trials, including the live and prerecorded video presentation conditions, underwent automated AU estimation using three software.

#### 2.6.1. FaceReader

FaceReader version 9 (Noldus Information Technology, Wageningen, The Netherland) is a commercial software that applies deep learning-based algorithms for face finding [47], facial modeling, and facial expression analysis [21]. The face modeling entailed a single-pass estimation of about 500 facial landmarks, which were then compressed via principal component analysis (PCA). FaceReader 9 analyzes 20 AUs (1, 2, 4, 5, 6, 7, 9, 10, 12, 14, 15, 17, 18, 20, 23, 24, 25, 26, 27, and 43) for the left and right unilateral AUs independently. AU classification was validated using the Amsterdam Dynamic Facial Expression Set [48] and the Warsaw Set of Emotional Facial Expression Pictures [49] with F1 scores of 0.777 and 0.764, respectively. Noldus also reported that on average, the East Asian model performed approximately 10% better on Asian faces than the general model. Hence, this study used the East Asian model.

#### 2.6.2. OpenFace

OpenFace is an open-source software [41], and version 2.2.0 uses a multitask cascaded convolutional neural network-based face detector [50] trained on the WIDER FACE [51] and CelebA [52] datasets. A convolutional experts’ constrained local model was used for landmark detection and tracking [53], in which a point distribution model captured landmark shape variation and patch experts model local appearance variations in each landmark. OpenFace 2.0 estimates the intensity of 17 AUs (1, 2, 4, 5, 6, 7, 9, 10, 12, 14, 15, 17, 20, 23, 25, 26, and 45). Facial appearance features were extracted using histograms of oriented gradients (HOG), and dimensionality was reduced using PCA. Person-specific neutral expression was extracted by computing the median value of face descriptors in the video sequence of a person, assuming that most frames contained neutral expressions. The extracted median face was subtracted from the feature descriptor, leading to a normalized feature. The normalized feature vector described the dynamic change from the neutral expression. The AU recognition framework uses linear kernel support vector machines for AU occurrence detection and support vector regression for AU intensity estimation [54], trained on the DISFA [55], SEMAINE [56], BP4D [57], UNBC-McMaster [58], Bosphorus [59], and FERA 2011 [60] datasets. AU classification has been validated with the DISFA dataset [55] and reported a mean Pearson correlation coefficient of 0.59 across 12 AUs (1, 2, 4, 5, 6, 9, 12, 15, 17, 20, 25, and 26; AU4: *r* = 0.70 and AU12: *r* = 0.85).

#### 2.6.3. Py-Feat

Py-Feat is an open-source Python toolbox [42]. Version 0.6.0 includes various pretrained models for each face-processing step. This study used default models. RetinaFace, a toolbox that performs 2D face alignment and 3D face reconstruction in a single shot, was used for face detection. In RetinaFace, the feature pyramid network obtains the input face images and outputs five feature maps of different scales, computed from the output of the corresponding ResNet residual stage (pretrained classification network on the ImageNet-11k dataset). Cascade multitask loss was then calculated, incorporating loss of face classification, box regression, five-facial-landmark regression, and 1k 3D vertices regression. RetinaFace was trained with WIDER FACE [51] with five manually annotated facial landmarks, AFLW [61], and an FDDB image set [62]. MobileFaceNets [63] was used for facial landmark detection, adopting a global depthwise convolution layer to output a discriminative feature vector, in place of the global average pooling layer, after the last non-global convolutional layer of a face feature embedding convolutional neural network. MobileFaceNets was trained on CIASIA-Webface [64] and validated using the Labeled Faces in the Wild [65] and AgeDB databases [66] with 99.28% and 93.05% accuracy, respectively. An XGBoost classifier, Feat-XGB [67], was used as the AU detector, which used PCA-reduced HOG features for AU predictions, as with OpenFace [54]. It was trained for 20 AUs (1, 2, 4, 5, 6, 7, 9, 10, 12, 14, 15, 17, 18, 20, 23, 24, 25, 26, 28, 43) using BP4D [57], BP4D+ [68], DISFA [55], DISFA+ [69], CK+ [70], JAFFE [71], Shoulder Pain [58], and EmotioNet [72] and validated using WIDER FACE [51], 300W [73], NAMBA [24], and BIWI-Kinect [74]. The average F1 score was 0.54 (AU4 = 0.64 and AU12 = 0.83) [42].

### 2.7. Statistical Analysis

#### 2.7.1. AU Response Accuracy

Trial-wise EMG response was defined as the mean differences in EMG natural log-transformed oscillation amplitudes between the neutral (0–1 s after stimulus onset) and maximal phases (2.5–3.0 s after stimulus onset) of the dynamic facial expression. AU response was analogously defined as the mean differences in AU estimation between the neutral (0–1 s after stimulus onset) and maximal phases (2.5–3.0 s after stimulus onset) of the dynamic facial expression. Using the sign (positive vs. negative) of the trial-wise EMG response as the ground truth, we calculated the proportion of AU4 and AU12 responses in the same sign as the EMG responses. OpenFace had an output of 0 for the whole AU12 time series in 680 positive and 680 negative trials out of 2996 trials (45.4%). Thus, we calculated the proportion of congruent OpenFace AU12 responses including and excluding those problematic trials.

#### 2.7.2. Spontaneous Mimicry Detection Accuracy

A mimicking response was defined as a context-congruent facial muscular or AU response. Under the positive conditions, we defined the ground truth of the mimicking response as ZM contraction and CS relaxation [6]. A true positive automated AU detection of mimicking responses constitutes a positive AU12 response and a null or negative AU4 response. In contrast, a false negative entails a null or negative AU12 response and a positive AU4 response. If no muscular mimicking response is observed, the false positive includes a positive AU12 response and a null or negative AU4 response. In contrast, a true negative constitutes a null or negative AU12 response and a positive AU4 response. Recall (sensitivity), specificity, precision (positive predictive value), negative predictive value, and F1 score were calculated to evaluate automated mimicry detection performance. OpenFace AU12 responses were estimated, both including and excluding trials of null AU12 amplitude. AU response and mimicry detection accuracy were analyzed using MATLAB R2023b. Because the automated AU4 estimation was affected by the placement of electrodes and fixative tapes (see Section 3.1), AU4 responses and time series were excluded from further analysis.

#### 2.7.3. Correlation between ZM and AU12 Responses

The R software (version 4.3.1; R Core Team, Vienna, Austria) and the package rmcorr 0.6.0 [75] were used to estimate the repeated measure correlation between ZM and AU12 responses. The rmcorr package models a random intercept for each individual and a fixed effect slope. This study estimated 95% confidence intervals using 10,000 bootstrapping, which does not require distributional assumptions.

#### 2.7.4. Detection of Emotional and Presentation Condition Interactions Using AU12 Responses

The R packages lme4 1.1-34, lmerTest 3.1-3, HLMdiag 0.5.0, and emmeans 1.8.8 were used along with the optimizer BOBYQA to perform linear mixed-effect (LME) model analysis as in Hsu et al., 2020 [39]. The dependent variables included the AU12 responses estimated via FaceReader, OpenFace, and Py-Feat. For each LME model, the emotional condition, presentation condition, and interactions were fixed effects, whereas “subject” was the random factor. The reference levels were “Negative” for the emotional condition and “Video” for the presentation condition. The parsimonious approach was adopted to stepwise increase model complexity until the more complex model was no longer significantly superior to the simpler one under the model comparison or when the model resulted in a singular fit. The most complex model that showed convergence was selected [76]. Initially, random intercepts for participants were included; the inclusion of by-subject random slopes for the effects of the emotional condition, presentation condition, their interaction term, and random intercepts for the study type depended on the model comparison results. Considering that the homogeneity of residual variance is an a priori assumption when seeking valid inferences from LME models [77,78], then upward residual and influence analyses were performed using the R package HLMdiag [79]. Trials were excluded if they produced an absolute standardized residual larger than 3 or a Cook’s distance greater than 1.5 × the interquartile range above the third quartile (*Q*_3_ + 1.5 × *IQR* as implemented in the HLMdiag package). Afterward, the second-level Cook’s distance was checked, as suggested in the HLMdiag package [79], and highly influential participants were excluded.

#### 2.7.5. Visualization of Condition-Wise ZM and AU12 Time Series Per Trial

The ZM time series were resampled to 29.97 Hz and 90 values per trial in MATLAB for comparable visualization of the EMG time series with the AU estimations. AU estimates with FaceReader quality scores lower than 0.6, Py-Feat FaceScores lower than 0.75, and OpenFace face recognition confidence scores lower than 0.75 were changed to NaNs, and these trials were excluded. Trials with OpenFace AU12 time series of all zeros were also excluded. The group-average amplitudes per condition and frame of the ZM and automated FACS AU12 were then calculated. The time series per measurement type and condition was shifted so that the first value of the time series per trial started at 0. The R packages ggplot2 3.4.3 and cowplot 1.1.1 were used to plot the group-average time series against the index of frames.

#### 2.7.6. Cross-Correlation between ZM and Automated FACS AU12 Time Series

The crosscorr function in MATLAB was used to calculate trial-wise cross-correlations between the resampled participant ZM and the automated FACS AU12 time series up to 45 lags (45 frames in 30 fps equals 1.5 s) during positive trials. The range of cross-correlation lags was consistent with previous findings in which spontaneous facial mimicry to static facial photos or prerecorded dynamic facial expressions showed a latency between 0.5 and 0.9 s [6,15]. The R package tidyverse 2.0.0 was then used to calculate condition-wise (live vs. video) mean cross-correlations per participant and group condition-wise (live vs. video) mean cross-correlations. The R package ggplot2 was implemented to visualize the cross-correlations from lag −45 to lag 45.

## 3. Results

This section presents the results in sequence of accuracy by AU response, ROC characteristics of spontaneous mimicry detection, the ability to detect the modulatory effect of the live condition, visualization of the EMG and automated FACS AU time series, and the cross-correlation for AU latency in relation to EMG measurements.

### 3.1. AU Response and Spontaneous Mimicry Detection Accuracy

In general, response and mimicry detection for AU4 was at the chance level (Table 2). The placement of the electrodes and fixative tapes affected the automated AU4 estimation, so AU4 responses and time series were excluded from further analysis.

FaceReader estimations for bilateral, left, and right AU12 suggested the negative influence of electrode placement on the right side of the participants’ faces. FaceReader bilateral AU12 estimation achieved the highest response accuracy at 0.564, followed by the FaceReader left AU12 estimation at 0.560 and the Py-Feat AU12 estimation at 0.554. After excluding 45.4% of all trials in which AU12 outputs were all 0, OpenFace showed an AU12 response detection accuracy of 0.556. Regarding AU12 mimicry detection, Py-Feat achieved the highest F1 score at 0.647, followed by FaceReader left AU12 estimation at 0.625 and FaceReader bilateral AU12 estimation at 0.618. After excluding trials where AU12 outputs were all 0, OpenFace showed a high F1 score of 0.727, while specificity was 0.235.

### 3.2. Repeated Measure Correlation between ZM and AU12 Responses

AU12 responses estimated that using all automated FACSs showed significant correlations with ZM responses (Table 3). The bilateral estimation by FaceReader showed the highest correlation with EMG-measured ZM responses, with the 95% CI lower bound higher than the 95% CI higher bound of the Py-Feat-estimated AU12 responses. OpenFace, when excluding all trials with 0 output for AU12, could show a correlation comparable to that of the FaceReader bilateral estimation.

### 3.3. Detection of Emotional and Presentation Condition Interactions Using AU12 Responses

AU12 responses estimated that using all automated FACSs could replicate our previous finding that the live condition enhanced spontaneous ZM mimicry [39,40]. The final model for ZM responses included the random intercept and random slope for the emotional conditions by the random factor “subject”. After model diagnostics, 2384 trials from 45 participants remained in the analysis. A significant interaction between the emotional and presentation conditions was observed. Simple-effect analysis according to the emotional condition showed that ZM responses under the positive-live condition were stronger than those under the positive-video condition; meanwhile, ZM responses under the negative-live condition were no different from those under the negative-video condition. Simple-effect analysis according to the presentation condition showed that ZM responses under the positive-live condition were stronger than those under the negative-live condition, while ZM responses under the positive-video condition were stronger than those under the negative-video condition (Table 4)

With regard to the FaceReader bilateral estimation, the final model included the random intercept and random slope for the emotional conditions by the random factor “subject”. After model diagnostics, 2360 trials from 45 participants were retained for analysis. A significant interaction was found between the emotional and presentation conditions. Simple-effect analysis showed the same pattern as in the ZM response model except that AU12 responses under the positive-video condition were no different from those under the negative-video condition (Table 4).

For Py-Feat estimation, the final model included the random intercept by the random factor “subject”. After model diagnostics, 2610 trials from 47 participants remained in the analysis. A significant interaction was observed between the emotional and presentation conditions. Simple-effect analysis showed the same pattern as in the ZM response model (Table 4).

Regarding OpenFace estimation, the final model included the random intercept and random slope for the emotional conditions by the random factor “subject”. After model diagnostics, 2389 trials from 45 participants remained in the analysis. A significant interaction was noted between the emotional and presentation conditions. Simple-effect analysis showed the same pattern as in the ZM response model, except AU12 responses under the negative-live condition were stronger than those under the negative-video condition, and AU12 responses under the positive-video condition were no different from those under the negative-video condition (Table 4).

### 3.4. Visualization of Condition-Wise ZM and AU12 Time Series

Figure 1 shows the group average of time series of amplitude per measurement type along each frame at 29.97 Hz. The ZM (Figure 1A) and all automated FACSs (Figure 1B–D) showed the pattern in which, toward the end of the trial, the positive-live condition evoked a stronger AU12 response than the positive-video condition. Concurrently, the AU12 remained around 0 in negative-live and negative-video conditions, as statistically demonstrated in Table 2. Visual inspection confirmed that the temporal pattern of the automated FACS AU12 is comparable to the standard ZM measurement. In Figure 1B, the FaceReader AU12 time series appeared much smoother and less noisy than all other measurement types. Figure 1D showed that a positive drift was observed in the OpenFace AU12 measurement, especially in negative-live and negative-video conditions, corresponding to a positive response in negative conditions, as statistically shown in Table 2.

### 3.5. Cross-Correlation between ZM and Automated FACS AU12 Time Series

The cross-correlation indicated the latency of the automated FACS-detected spontaneous facial mimicry compared to ZM EMG in the positive conditions. When cross-correlated with the EMG time series, the FaceReader AU12 time series reached peak cross-correlation with delays between 167 and 300 ms (lag 5 and 9, *r* = 0.093) relative to the ZM in the positive-live condition and between 200 and 333 ms (lag 6 and 10, *r* = 0.066) relative to the ZM in the positive-video condition (Figure 2, red). The cross-correlation curve appeared much smoother than other automated FACSs. The Py-Feat AU12 time series reached peak cross-correlation with delays between 67 and 100 ms (lag 2 and 3, *r* = 0.074) relative to the ZM in the positive-live condition and between 100 and 133 ms (lag 3 and 4, *r* = 0.057) relative to the ZM in the positive-video condition (Figure 2, purple). The OpenFace AU12 time series reached peak cross-correlation with delays of 33 ms (lag 1, *r* = 0.139) relative to the ZM in the positive-live condition and between 67 and 100 ms (lag 2 and 3, *r* = 0.114) relative to the ZM in the positive-video condition (Figure 2, green). The OpenFace results excluded trials in which all OpenFace AU12 outputs were zero. Generally, the automated FACS showed a slight delay compared to facial EMG recordings. Latency might be smaller in the positive-live than in the positive-video condition, but this could not be statistically tested.

## 4. Discussion

Simultaneous facial EMG and frontal facial video recording were used to validate the performance of the automated FACS software detection of spontaneous facial mimicry. For AU12 mimicry detection, FaceReader 9 and Py-Feat could achieve F1 scores above 0.6 (Table 2). All AU12 response estimations were significantly correlated with ZM response, capturing up to about 25% variance of ZM recordings (Table 3). All AU12 response estimations could replicate the previous finding that live interaction enhanced smile mimicry in the positive conditions (Table 4), as visualized in the group-average time series (Figure 1). All AU12 time series showed slight delays (around 100 ms) compared to the ZM time series (Figure 2), indicating that frontal facial video recordings were much less precise and robust than facial EMG in facial mimicry detection and could not replace facial EMG. However, the automated FACS could still detect subtle differences in spontaneous facial mimicry and could be valuable for this purpose while being much less disruptive to participants’ visual appearance and comfort. In addition, the automated FACS analysis of video recordings may be more suitable for detecting facial mimicry in the wild. Compared to previous studies (Table 1), we provided clear quantitative analyses of the ROC characteristic matrices regarding the automated FACS performance of spontaneous facial mimicry detection, the automated FACS ability to test the modulation of live effects on spontaneous facial mimicry, and the latency of the automated FACS time series compared with EMG measurements.

The automated FACS was highly susceptible to the visual blockade of facial parts. Because of how electrodes and fixative tapes were placed on the eyebrows for CS recording, the performance of AU4 estimation was at the chance level and only ZM/AU12 estimation could be reliably evaluated. FaceReader 9.0 also provided evidence that the side of the face with the electrode attachment for ZM (participant’s left face, which was the right side in the video recordings) demonstrated lower sensitivity, positive predictive value, and F1 score than the side without the electrode. This showed that in real-life applications, the automated FACS would not be robust against the presence of facial accessories or scars in addition to the effect of ever-changing viewpoints [24]. FaceReader, which could evaluate bilateral AUs separately, seems more useful against partial visual blockade. Hence, future research efforts could assess the performance of automated FACSs, especially those with facial symmetry assumption (e.g., OpenFace and Py-Feat), on faces with anomalies [80].

FaceReader 9.0 has some advantages. Its ability to evaluate bilateral AUs separately provided a more realistic estimation since the face and facial actions are often asymmetric [81]. Bilateral estimation might have also contributed to the robustness of AU estimation against partial facial blockade. The East Asian module might have also enhanced the AU estimation performance in the current participant population, that is, Japanese females. However, without bilateral estimation and the East Asian module, Py-Feat achieved a comparable or even higher F1 score in smile mimicry detection than FaceReader. The advantages seemed to have contributed to improved performance in detecting spontaneous facial mimicry in this study compared with the performance of FaceReader 7.0 used in Höfling and colleagues (2021) in which spontaneous facial mimicry during passive viewing was only detected by the EMG delta but not the FaceReader valence output [34]. However, based on time series and cross-correlation visualization (Figure 1 and Figure 2), FaceReader appeared to demonstrate temporal smoothing across frame-wise estimations to reduce temporal noise, resulting in smooth AU time series compared to other automated FACS software and possibly introduced some latency in AU estimation compared to OpenFace and Py-Feat.

Some unexpected phenomena were observed in OpenFace. OpenFace AU12 output was completely zero in 45.4% of the trials despite a confidence score of above 0.75 and the estimation of other AUs appearing normal, and the issue was replicated despite repeated OpenFace estimations. After excluding such problematic trials, which included about half of the data, OpenFace showed a high sensitivity and F1 score, and low specificity in smile mimicry detection, rendering an unbalanced performance and a lack of robustness. Furthermore, the OpenFace AU12 time series showed a positive drift, resulting in positive responses of AU12 during negative trials, which was not observed in ZM and other automated FACSs. Readers should observe caution when interpreting OpenFace AU12 estimation for video recordings.

Using the current XGBoost classifier, Py-Feat’s smile mimicry detection performance was comparable to that of FaceReader based on the receiver operating characteristics (Table 2). In addition, the Py-Feat AU12 time series visualization did not show smoothing as with FaceReader. While the current default modules performed comparably well, the impact of module change with Py-Feat updates was also noted. For example, the initial analysis in this study used an earlier version of Py-Feat with the *logistic* AU model, which was dropped in version 0.5.0. Reanalysis using the new XGBoost AU model vastly improved smile mimicry detection performance, but like any software, performance requires a reevaluation with major updates. Hopefully, the EMG validation of AU models could be a benchmark used by developers in the future. Furthermore, algorithms involving machine learning classifications might improve the performance of the EMG and FACS analysis of spontaneous facial mimicry [82,83,84,85].

Our results indicated that the automated FACS, despite its ability to detect spontaneous facial mimicry and the modulatory effect exerted by the live condition [44,45,46], is far from being as sensitive and accurate as facial EMG for such tasks. This raises an issue when it is used to test clinical populations. Its lack of sensitivity and accuracy might render it more challenging to detect spontaneous facial mimicry of smaller effect sizes among specific clinical populations, such as individuals with autism [86] or schizophrenia [87] and may generate null results. In contrast, a reduced effect should have been distinguished from the absence of an effect, and the exact nature of altered responses should have been revealed. Researchers should be careful when interpreting inferences about facial communication in clinical or subclinical populations based on the automated FACS.

This study has several limitations. First, it needed to be fully optimized for facial video analysis. Electrode size and the extensive use of fixative tapes negatively influenced the validation of AU4 estimation using CS recordings. This issue might be addressed by smaller and lighter electrodes used in recently developed wearable EMG devices [88]. Second, it might have been too stringent to use facial mimicry detection as the sole benchmark for evaluating automated FACS performance. Spontaneous facial mimicry is known to have small amplitudes and is not always visually detectable [31,32]. Therefore, it was expected that the automated FACS based on facial recordings could not be as sensitive as facial EMG. However, given the increasing number of researchers using automated FACS to detect spontaneous facial mimicry [35,36,37,38], it is first essential to evaluate whether the software performs as well as the researchers’ expectation. Third, the present study only evaluated the pairs of CS–AU4 and ZM–AU12. CS and ZM are the most standard and frequently measured muscles in spontaneous facial mimicry studies [89]. However, it would be valuable for future studies to validate other muscle–AU pairs contributing most to the discrete emotional facial expressions. This could further improve emotional facial expression classification based on multivariate AU estimates. As the EMG measurement itself could interfere with facial video recordings for automated FACS analysis, the number of facial muscles that could be reliably recorded while maintaining the visual intactness of the facial appearance is limited.

In conclusion, facial EMG was used as a standard to validate spontaneous facial mimicry detection when viewing live and prerecorded dynamic facial expressions. The automated FACS was found to be less sensitive and less accurate than facial EMG despite being similarly capable of detecting comparable mimicry responses. Each software has its advantages and problems, and researchers should be cautious when examining output from the automated FACS. Developers should also consider the EMG validation of AU estimation from the automated FACS as a benchmark.

## Figures and Tables

**Figure 1 sensors-23-09076-f001:**
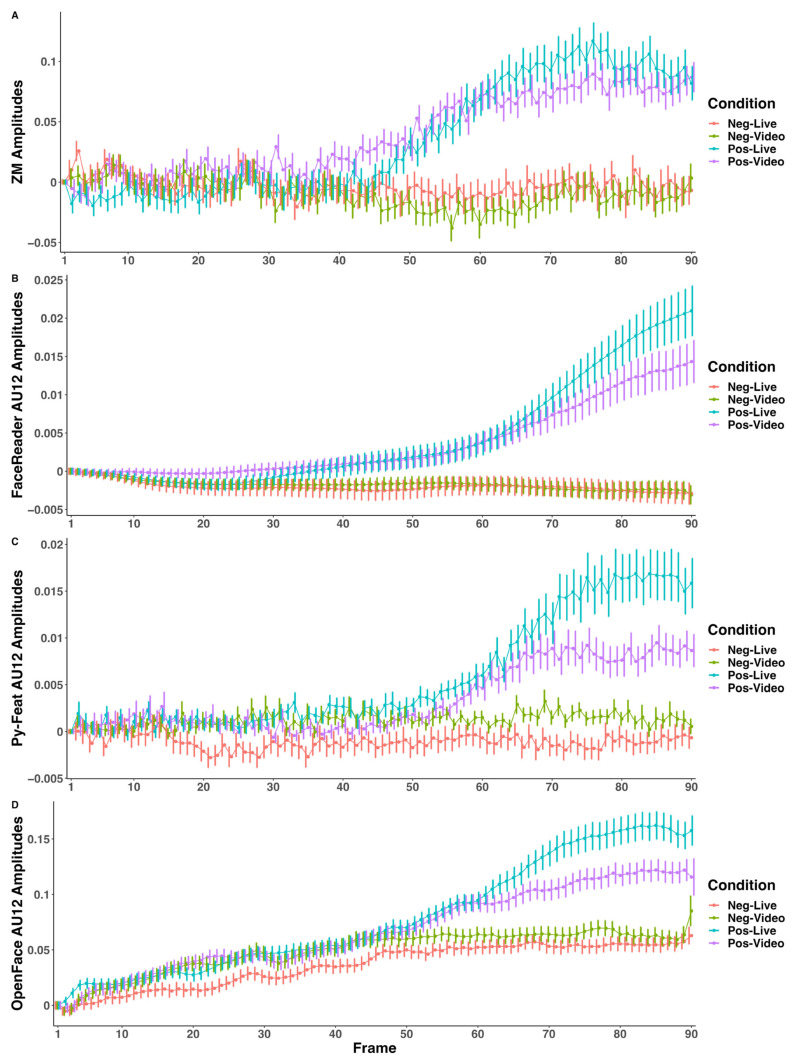
Condition-wise group-average EMG and automated AU12 estimation time series. Group-average amplitude was calculated for each condition, measurement type, and frame along each trial. Panel (**A**): EMG ZM measurements. Panel (**B**): FaceReader AU12 estimations. Panel (**C**): Py-Feat AU12 estimations. Panel (**D**): OpenFace AU12 estimations. The time series per trial was shifted so that the first value of each trial started at 0. The time series per measurement type and condition were each connected with lines. The error bar indicates the standard error. The *x*-axis indicates the index of frames from 1 to 90. At 29.97 fps, the between-frame interval was about 67 ms. The *y*-axis shows the group-average amplitude.

**Figure 2 sensors-23-09076-f002:**
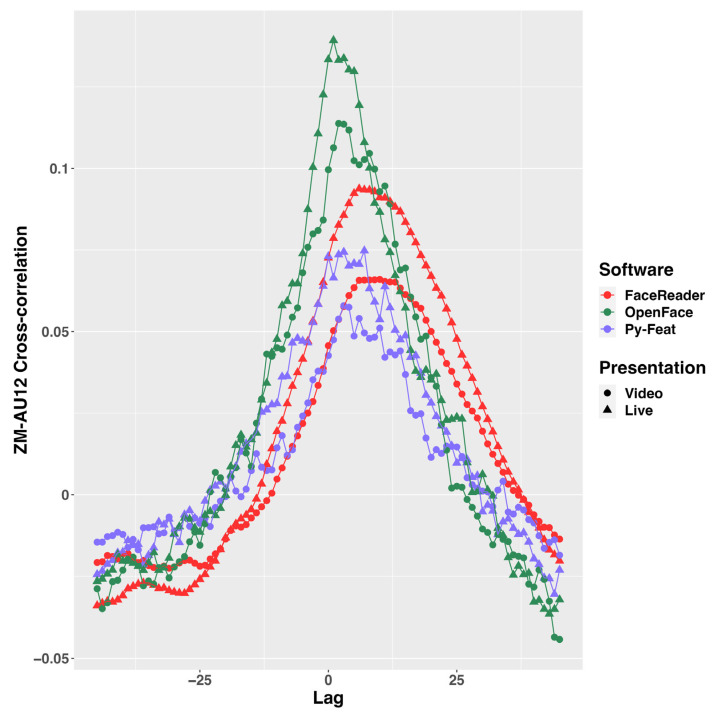
Group-average cross-correlation of automated AU12 estimations with ZM time series in the positive conditions. The *x*-axis indicates the lag in frames (67 ms per frame). A positive lag (peak cross-correlation occurred after 0) showed that the automated FACS AU12 was slower than the ZM recording.

**Table 1 sensors-23-09076-t001:** Comparison of studies validating automated FACS AU estimation using EMG.

Authors	Software	Measurements	Task	FACS-EMG Comparisons
D’Arcey (2013) [27]	FaceReader 4.0	FACS: happy, angry, disgusted, and sad. EMG: CS and ZM.	Volitional mimicry.	EMG–emotion intensity correlation.
Beringer et al. (2019) [28]	FACET 6.1.2667.3	FACS: AU4 and AU12. EMG: CS and ZM.	Volitional facial expressions.	ROC characteristics. Comparison of priming effects on the reaction time. Visualization of time series.
Höfling et al. (2020) [33]	FaceReader 7.0	FACS: valence, arousal, AU4, and AU12. EMG: CS and ZM.	Passive viewing (spontaneous facial mimicry).	Separate ANOVA for EMG and FACS outputs. Visualization of time series.
Höfling et al. (2021) [34]	FaceReader 7.0	FACS: valence, arousal. EMG: CS and ZM.	Volitional mimicry. Passive viewing (spontaneous facial mimicry). Mimicry inhibition.	Separate ANOVA for EMG and FACS outputs. Visualization of time series.
Kulke et al. (2020) [30]	AFFDEX 1.0	FACS: AU4 and AU 12. EMG: CS, ZM, and orbicularis oculi.	Volitional mimicry.	Correlation between the difference of the FACS estimations and EMG measurements.
The present study	FaceReader 9.0 OpenFace 2.2.0 Py-Feat 0.6.0	FACS: AU4 and AU 12. EMG: CS and ZM.	Passive viewing (spontaneous facial mimicry). Live effect vs. prerecorded videos.	ROC characteristics. Correlation between the difference of the FACS estimations and EMG measurements. Separate LME analysis for EMG and FACS outputs to test the live effect. Cross-correlation to test latency. Visualization of time series.

**Table 2 sensors-23-09076-t002:** Performance of automated FACS in spontaneous facial response and mimicry detection.

	AU12					AU4				
	Res. Acc	Mimicry				Res. Acc	Mimicry			
		Sensitivity	Specificity	PPV	F1		Sensitivity	Specificity	PPV	F1
**FRB**	0.564	0.624	0.482	0.612	0.618	0.507	0.526	0.493	0.497	0.511
**FRL**	0.560	0.627	0.501	0.622	0.625	0.507	0.529	0.492	0.497	0.513
**FRR**	0.534	0.557	0.488	0.588	0.572	0.501	0.519	0.488	0.491	0.505
**PF**	0.554	0.668	0.481	0.628	0.647	0.486	0.529	0.438	0.473	0.499
**OF**	0.288	0.475	0.638	0.632	0.543	0.480	0.475	0.523	0.487	0.481
**OF***	0.556	0.856	0.235	0.632	0.727					

Abbreviations. AU: action unit; FRB: FaceReader bilateral estimation; FRL: FaceReader left face estimation; FRR: FaceReader right face estimation; OF: OpenFace with all data; OF*: OpenFace excluding trials where all AU12 outputs were 0; PF: Py-Feat estimation; PPV: positive predictive value; Res. Acc: response accuracy.

**Table 3 sensors-23-09076-t003:** Repeated measure correlation between ZM and AU12 responses by different automated FACS.

	*r*	df	95% CI	*p*
**FRB**	0.505	2945	(0.431, 0.573)	1.095 × 10^−190^
**FRL**	0.360	2945	(0.260, 0.452)	1.026 × 10^−90^
**FRR**	0.471	2945	(0.399, 0.539)	2.338 × 10^−162^
**PF**	0.338	2945	(0.246, 0.423)	1.999 × 10^−79^
**OF**	0.445	2945	(0.367, 0.512)	2.460 × 10^−143^
**OF***	0.517	1588	(0.433, 0.589)	2.788 × 10^−109^

Abbreviations. df: degrees of freedom. Others: see Table 2 footnotes.

**Table 4 sensors-23-09076-t004:** Fixed effect of emotional and presentation conditions interaction as shown by automated FACS.

	Mean	*SE*	df	Lower CI	Upper CI	*t*	*p*
**EMG ZM Responses**							
**Intercept**	5.242 × 10^−3^	5.500 × 10^−3^	43.99	−5.640 × 10^−3^	0.01614	0.953	0.3458
**Emotion**	2.601 × 10^−2^	7.362 × 10^−3^	43.98	0.01145	0.04061	3.532	<0.0001 *
**Presentation**	5.262 × 10^−3^	2.113 × 10^−3^	2296	0.01120	0.00940	2.490	0.0128 *
**E*P**	5.917 × 10^−3^	2.989 × 10^−3^	2296	5.507 × 10^−5^	0.01177	1.980	0.0478 *
** *Simple effects according to the emotional condition* **							
**PL**	0.0303	0.0102	48.3	0.00989	0.05073		
**PV**	0.0170	0.0101	47.8	−0.00343	0.03733		
**PL-PV**	0.0134	0.0042	2293			3.167	0.0016 *
**NL**	−0.0124	0.0046	71.8	−0.02147	−0.00330		
**NV**	−0.0139	0.0045	71.0	−0.02296	−0.00485		
**NL-NV**	0.0015	0.0042	2298			0.360	0.7190
** *Simple effects according to the presentation condition* **							
**PL-NL**	0.0427	0.0108	51.8			3.936	0.0002 *
**PV-NV**	0.0309	0.0108	51.2			2.853	0.0062 *
**FaceReader Bilateral AU12**							
**Intercept**	7.592 × 10^−4^	5.601 × 10^−4^	43.99	−0.00035	0.00187	1.355	0.1822
**Emotion**	1.779 × 10^−3^	6.263 × 10^−4^	43.97	0.00054	0.00303	2.840	0.0068 *
**Presentation**	4.607 × 10^−4^	1.261 × 10^−4^	2273	0.00021	0.00071	3.654	0.0003 *
**E*P**	7.971 × 10^−4^	1.783 × 10^−4^	2273	0.00045	0.00115	4.470	<0.0001 *
** *Simple effects according to the emotional condition* **							
**PL**	0.0027	0.0010	45.6	0.00079	0.00470		
**PV**	0.0013	0.0010	45.5	−0.00066	0.00325		
**PL-PV**	0.0014	0.0003	2270			5.729	<0.0001 *
**NL**	−0.0006	0.0003	59.8	−0.0012	0.00009		
**NV**	−0.0004	0.0003	59.3	−0.0011	0.00024		
**NL-NV**	−0.0001	0.0003	2276			0.579	0.5629
** *Simple effects according to the presentation condition* **							
**PL-NL**	0.0033	0.0009	47.7			−3.664	0.0006 *
**PV-NV**	0.0017	0.0009	47.5			−1.903	0.0631
**Py-Feat AU12**							
**Intercept**	1.117 × 10^−3^	3.553 × 10^−4^	46.00	0.00042	0.00182	3.145	0.0029 *
**Emotion**	1.271 × 10^−3^	1.394 × 10^−4^	2563	0.00100	0.00154	9.115	<0.0001 *
**Presentation**	6.478 × 10^−4^	1.393 × 10^−4^	2562	0.00037	0.00092	4.651	<0.0001 *
**E*P**	7.813 × 10^−4^	1.968 × 10^−4^	2561	0.00040	0.00117	3.969	<0.0001 *
** *Simple effects according to the emotional condition* **							
**PL**	0.0029	0.0004	71.9	0.00207	0.00366		
**PV**	0.0012	0.0004	68.8	0.00038	0.00195		
**PL-PV**	0.0017	0.0003	2561			6.035	<0.0001 *
**NL**	0.00029	0.0004	69.0	−0.00050	0.00107		
**NV**	0.00015	0.0004	68.6	−0.00063	0.00094		
**NL-NV**	0.00014	0.0003	2561			0.489	0.6248
** *Simple effects according to the presentation condition* **							
**PL-NL**	0.0026	0.0003	2563			9.146	<0.0001 *
**PV-NV**	0.0001	0.0003	2561			3.691	0.0002 *
**OpenFace AU12**							
**Intercept**	1.550 × 10^−2^	4.261 × 10^−3^	42.78	0.00707	0.02395	3.637	0.0007 *
**Emotion**	4.797 × 10^−3^	1.589 × 10^−3^	35.54	0.00167	0.00799	3.019	0.0047 *
**Presentation**	3.628 × 10^−3^	5.753 × 10^−4^	2292	0.00250	0.00476	6.306	<0.0001 *
**E*P**	2.178 × 10^−3^	8.136 × 10^−4^	2292	0.00058	0.00377	2.677	0.0075 *
** *Simple effects according to the emotional condition* **							
**PL**	0.0225	0.0051	45.2	0.01228	0.0328		
**PV**	0.0152	0.0051	45.1	0.00498	0.0255		
**PL-PV**	0.0073	0.0012	2300			6.362	<0.0001 *
**NL**	0.0136	0.0037	46.2	0.00617	0.0210		
**NV**	0.0106	0.0037	46.2	0.00322	0.0180		
**NL-NV**	0.0030	0.0012	2299			2.562	0.0105 *
** *Simple effects according to the presentation condition* **							
**PL-NL**	0.0090	0.0024	56.1			3.746	0.0004 *
**PV-NV**	0.0046	0.0024	55.9			1.928	0.0589

Notes. * *p* < 0.05. Formula for ZM, FaceReader and OpenFace: ZM/AU12~1 + emotional_condition * presentation_condition + (1 + emotional_condition | subject). Formula for Py-Feat: AU12~1 + emotional_condition * presentation_condition + (1 | subject). Abbreviations. df: degrees of freedom; EMG: electromyography; NL: negative-live condition; NV: negative-video condition; PL: positive-live condition; PV: positive-video condition; SE: standard error; CI: confidence interval; ZM: zygomaticus major.

## Data Availability

The data and scripts presented in this study are openly available on OSF at https://osf.io/3gcxj/.

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
