# Peer review of "Electromyographic Validation of Spontaneous Facial Mimicry Detection Using Automated Facial Action Coding"

_sensors, 2023, doi:10.3390/s23229076_

Round 1

Reviewer 1 Report

Comments and Suggestions for Authors

The authors present the article entitled “Electromyographic Validation of Facial Mimicry Detection Using Automated Facial Action Coding”

The article presents the following general concerns:

  • Add hyperlinks to tables, figures, and references.

  • At the end of the introduction, describe the structure of the text. 

  • This kind of text must be written in the third person or passive voice.

  • Furthermore, there is a clear lack of editing in the document.

  • In the introduction section, it is necessary to quantitatively and qualitatively analyze previous works.

  • The problem or controversy to be resolved concerning previous works is not clear. Please clarify

  • It is necessary to mention the main contributions of this work.

  • Add a brief introduction between sections and subsections.

  • The use of tables is recommended for the materials and methods section to provide relevant information.

  • In the comparison section, please add a comparative table of the results of this work with previous works.

  • I recommend to reach the EMG signals background topic before line 154 in order to illustrate to the reader the facts in this matter, for instance, the following works can be considered to be referenced: Support vector machine-based emg signal classification techniques: a review; A study of computing zero crossing methods and an improved proposal for emg signals; A novel methodology for classifying emg movements based on svm and genetic algorithms; Optimizing emg classification through metaheuristic algorithms

  • Validate that all decimal values have zero before the decimal point. For example, Table 1 does not have this format.

  • Please center the tables.

  • Figure 1 requires better analysis and description.

  • Please update the references since more than 76% of the citations are less than 2018.

Comments on the Quality of English Language

The authors present the article entitled “Electromyographic Validation of Facial Mimicry Detection Using Automated Facial Action Coding”

The article presents the following general concerns:

  • Add hyperlinks to tables, figures, and references.

  • At the end of the introduction, describe the structure of the text. 

  • This kind of text must be written in the third person or passive voice.

  • Furthermore, there is a clear lack of editing in the document.

  • In the introduction section, it is necessary to quantitatively and qualitatively analyze previous works.

  • The problem or controversy to be resolved concerning previous works is not clear. Please clarify

  • It is necessary to mention the main contributions of this work.

  • Add a brief introduction between sections and subsections.

  • The use of tables is recommended for the materials and methods section to provide relevant information.

  • In the comparison section, please add a comparative table of the results of this work with previous works.

  • I recommend to reach the EMG signals background topic before line 154 in order to illustrate to the reader the facts in this matter, for instance, the following works can be considered to be referenced: Support vector machine-based emg signal classification techniques: a review; A study of computing zero crossing methods and an improved proposal for emg signals; A novel methodology for classifying emg movements based on svm and genetic algorithms; Optimizing emg classification through metaheuristic algorithms

  • Validate that all decimal values have zero before the decimal point. For example, Table 1 does not have this format.

  • Please center the tables.

  • Figure 1 requires better analysis and description.

  • Please update the references since more than 76% of the citations are less than 2018.

Author Response

Thank you for your valuable and constructive comments on our manuscript. We have carefully revised the manuscript according to your suggestions. All revisions were marked using the “Track Changes” function.

The authors present the article entitled “Electromyographic Validation of Facial Mimicry Detection Using Automated Facial Action Coding”

The article presents the following general concerns:

  1. Add hyperlinks to tables, figures, and references.

Authors’ response:

This manuscript was made with the Word Template provided by the journal, and the hyperlinks will be added during the production phase for publication if accepted.

  1. At the end of the introduction, describe the structure of the text.

Authors’ response:

We have added a paragraph at the end of the introduction to describe the manuscript's structure.

  1. This kind of text must be written in the third person or passive voice.

Authors’ response:

For stylish consistency, we followed the suggestion to change all active voice sentences to either third person or passive voice.

  1. Furthermore, there is a clear lack of editing in the document.

Authors’ response:

A professional English-language editing service (https://www.enago.jp/) has made language-related changes to the original and revised manuscript.

  1. In the introduction section, it is necessary to quantitatively and qualitatively analyze previous works.

Authors’ response:

We have enhanced the introduction to provide more details on previous studies on detecting spontaneous facial mimicry using automated FACS. However, comparable previous studies are scarce, while the ones tackling this issue did not provide much quantitative statistical analysis, and we have emphasized these points as the contribution and advantage of the present study.

  1. The problem or controversy to be resolved concerning previous works is not clear. Please clarify

Authors’ response: We have enhanced this information in the last second paragraph of the introduction, emphasizing the unsolved issues to be resolved in the present paper.

  1. It is necessary to mention the main contributions of this work.

Authors’ response:

We have enhanced this information in the last second paragraph of the introduction and the first paragraph of the discussion.

  1. Add a brief introduction between sections and subsections.

Authors’ response:

We have added a brief introduction between sections and subsections in the Methods and Material, and Results sections.

  1. The use of tables is recommended for the materials and methods section to provide relevant information.

Authors’ response:

While it is difficult to make procedural details of the materials and methods into tables, we have summarized the methodology of the present study in Table 1 in comparison with previous studies.

  1. In the comparison section, please add a comparative table of the results of this work with previous works.

Authors’ response:

As mentioned in reply to Reviewer 1’s point 4, previous studies did not perform identical quantitative statistical analysis as in the present study, results of the present study could not be compared directly to results of previous works. However, we have made a comparable table in the introduction (Table 1) to compare our approach with those of previous studies.

  1. I recommend to reach the EMG signals background topic before line 154 in order to illustrate to the reader the facts in this matter, for instance, the following works can be considered to be referenced: Support vector machine-based emg signal classification techniques: a review; A study of computing zero crossing methods and an improved proposal for emg signals; A novel methodology for classifying emg movements based on svm and genetic algorithms; Optimizing emg classification through metaheuristic algorithms

Authors’ response:

EMG signal classification might be relevant for future studies in spontaneous facial mimicry research, so we referred to the suggested works in the last second paragraph of the discussion as future directions.

  1. Validate that all decimal values have zero before the decimal point. For example, Table 1 does not have this format.

Authors’ response:

We noted the inconsistency in the styling and have followed the suggestion to add 0s before the decimal.

  1. Please center the tables.

Authors’ response:

The tables have been centered.

  1. Figure 1 requires better analysis and description.

Authors’ response:

Figure 1 serves for the purpose of visualization of time series and to inspect the output in a qualitative way. To improve readability, we have separated four measurements into four subplots, increased the spacing in the x-axis labels to avoid cluttering, and updated section 2.7.5. and Figure 1 legend.

  1. Please update the references since more than 76% of the citations are less than 2018.

Authors’ response:

We have enhanced the description of previous studies and cited more recent studies.

Reviewer 2 Report

Comments and Suggestions for Authors

Dear authors,

With interest we were reading the paper and we have the following critical comment:

1. We found a paper of Hofling presenting a similar topic using emotional expressions and not only AU's

Alison C. Hollanda*, Garret O’Connellb* and Isabel Dziobe. Facial mimicry, empathy, and emotion recognition: a meta-analysis ofcorrelation. COGNITION AND EMOTION2021, VOL. 35, NO. 1, 150–168https://doi.org/10.1080/02699931.2020.1815655

T. Tim A. Höfling, Georg W. Alpers, Antje B. M. Gerdes & Ulrich Föhl. Automatic facial coding versus electromyography of mimicked, passive, and inhibited facial response to emotional faces. Cognition and Emotion, 35:5, 874-889, DOI: 10.1080/02699931.2021.1902786

After clarification the paper should be considered again

 2. Electromyographic emotion assessment need sensors attached to the face. The camera based methods are non intrusive and could be used for emotion detection of faces in the wild

3. The authors doesn't state if they consider emotions of different intensities

4. It would be interesting if more AU's than only AU12 would be researched

5. The paper is well written and presentation of experimental results is very detailed.

6. The authors state that electromyographic methods outperforms the camera based methos but are limited in use and we were wondering hoe good the performance is (statistical measurements needed)

Author Response

Thank you for your valuable and constructive comments on our manuscript. We have carefully revised the manuscript according to your suggestions. All revisions were marked using the “Track Changes” function.

  1. We found a paper by Hofling presenting a similar topic using emotional expressions and not only AU's

Alison C. Hollanda*, Garret O’Connellb* and Isabel Dziobe. Facial mimicry, empathy, and emotion recognition: a meta-analysis ofcorrelation. COGNITION AND EMOTION2021, VOL. 35, NO. 1, 150–168https://doi.org/10.1080/02699931.2020.1815655

  1. Tim A. Höfling, Georg W. Alpers, Antje B. M. Gerdes & Ulrich FöhlAutomatic facial coding versus electromyography of mimicked, passive, and inhibited facial response to emotional facesCognition and Emotion, 35:5, 874-889, DOI: 10.1080/02699931.2021.1902786

After clarification the paper should be considered again.

Authors’ response:

We thank the reviewer for reminding us of the other publication by Höfling and colleagues.

Höfling et al. 2021 added the volitional mimicking and inhibition conditions in addition to the passive viewing condition. Note that spontaneous facial mimicry, which is subconscious with a small effect size and difficult to detect with human eyes, happens in the passive viewing condition. The effect of volitional mimicking is easy to detect. Moreover, this study focused only on the relationship between the FaceReader Valence output and the “EMG diff” (same as the EMG delta in Höfling et al. 2020), not the AU estimation per se.

Höfling et al. 2021 reported that the FaceReader Valence did not differentiate between emotion conditions during passive viewing, while the “EMG diff” could (Figure 2, middle panel). Their supplement B showed that isolated ZM and CS measurements did not detect spontaneous facial mimicry in the passive condition in their EMG recordings. In contrast, we showed robust spontaneous facial mimicry in EMG ZM, CS, and AU12 in automated FACS. In addition, we showed that the enhanced spontaneous mimicry in the live condition (audience effect), which was detected using EMG, could be detected in the AU12 estimation of automated FACS. This is absent in Höfling et al. 2020 and 2021, as they did not use live stimuli. In addition, we statistically examined how much variance in the EMG ZM response during passive viewing could be captured in the AU12 estimation of automated FACS using repeated measure correlations, which was not tested in Höfling et al. 2020 and 2021. In addition, we statistically examined the latency of spontaneous mimicry detected by AU12 of automated FACS compared to the ZM time series using cross-correlation, which was absent in Höfling et al. 2020 and 2021. Höfling et al. only validated FaceReader 7, while we used FaceReader 9, which incorporated many updates, including deep learning-based algorithms mentioned in Section 2.6.1. In addition, we evaluated OpenFace and Py-Feat. The findings of FaceReader 7 could not be generalized to newer versions and other automated FACS. The version difference might explain the improved ability to detect spontaneous facial mimicry during passive viewing in version 9. Including the open-source OpenFace and Py-Feat in the present manuscript would be informative for other researchers since open-source software is more readily available.

The abovementioned comparisons have been implemented in the introduction, including Table 1, to emphasize the contribution of the present work. We have also added the two mentioned articles in the reference.

  1. Electromyographic emotion assessment need sensors attached to the face. The camera based methods are non intrusive and could be used for emotion detection of faces in the wild

Authors’ response:

We have added this advantage of the camera-based method in the first paragraph of the discussion.

  1. The authors doesn't state if they consider emotions of different intensities

Authors’ response:

The models performed the positive and negative facial expressions with consistent intensity, and we have added this information to section 2.4, “prerecorded and live videos.” The valence and arousal level of perceived emotion and emotional contagion would vary among individuals, as shown in the valence and arousal ratings in our previous publication of the same paradigm (Hsu et al., 2020, 2022). 

  1. It would be interesting if more AU's than only AU12 would be researched

Authors’ response:

We agree with the reviewer it would be interesting to validate AUs other than AU12. However, we only collected the EMG measurements of CS and ZM in the present study, corresponding to AU4 and AU12. As we reported, AU4 estimation was adversely affected by the electrode placement. As shown in the meta-analysis Reviewer 2 has mentioned in point 1 (Holland et al., 2020), CS and ZM are the standard and most frequently measured muscles in spontaneous facial mimicry studies. Nevertheless, we have added this point as a limitation in the last second paragraph of the discussion.

  1. The paper is well written and presentation of experimental results is very detailed.

Authors’ response:

We thank the reviewer for the comment.

  1. The authors state that electromyographic methods outperforms the camera based methos but are limited in use and we were wondering hoe good the performance is (statistical measurements needed)

Authors’ response:

Studies of spontaneous facial mimicry during passive viewing, since the first mention by Dimberg (Dimberg, 1982), has always relied on EMG as the standard and golden measure for detection, also shown in the meta-analysis Reviewer2 mentioned in point 1 (Holland et al., 2020). Because spontaneous facial mimicry is not an intentional action, there is currently no better way than EMG to detect spontaneous facial mimicry; the performance of EMG could not be compared with a better standard statistically. The present study assumes that what EMG measured is the ground truth, which infers that the EMG is always correct. Under such an assumption, we tested how various automated FACS software performed.

Bibliography

Dimberg, U. (1982). Facial reactions to facial expressions. Psychophysiology, 19, 643–647. https://doi.org/10.1111/j.1469-8986.1982.tb02516.x

Holland, A. C., O’Connell, G., & Dziobek, I. (2020). Facial mimicry, empathy, and emotion recognition: A meta-analysis of correlations. Cognition and Emotion, 1–19. https://doi.org/10.1080/02699931.2020.1815655

Hsu, C.-T., Sato, W., Kochiyama, T., Nakai, R., Asano, K., Abe, N., & Yoshikawa, S. (2022). Enhanced mirror neuron network activity and effective connectivity during live interaction among female subjects. NeuroImage, 263(119655), 1–19. https://doi.org/10.1016/j.neuroimage.2022.119655

Hsu, C.-T., Sato, W., & Yoshikawa, S. (2020). Enhanced emotional and motor responses to live versus videotaped dynamic facial expressions. Scientific Reports, 10(1), Article 1. https://doi.org/10.1038/s41598-020-73826-2

Round 2

Reviewer 1 Report

Comments and Suggestions for Authors

The manuscript can be accepted 

Reviewer 2 Report

Comments and Suggestions for Authors

Dear authors,

Thanks for the major revision, we are satisfied with the corrections